# A mechanical model of ocular bulb vibrations and implications for acoustic tonometry

**Nicoletta Tambroni[1], Giuseppe Tomassetti[2]\*, Silvia Lombardi[3], Rodolfo Repetto[1]**

**1** Department of Civil, Chemical and Environmental Engineering, University of Genoa, Genoa, Italy,
**2** Department of Industrial, Electronic, and Mechanical Engineering, Roma Tre University, Rome, Italy, **3** Gran Sasso Science Institute, L'Aquila, Italy

\* giuseppe.tomassetti@uniroma3.it

**Data Availability Statement:** All relevant data are within the paper and its Supporting information files.

**Funding:** Giuseppe Tomassetti acknowledges support from the Italian Ministry of University and

## Abstract

In this study, we propose a comprehensive mechanical model of ocular bulb vibrations and discuss its implications for acoustic tonometry. The model describes the eye wall as a spherical, pre-stressed elastic shell containing a viscoelastic material and accounts for the interaction between the elastic corneoscleral shell and the viscoelastic vitreous humor. We investigate the natural frequencies of the system and the corresponding vibration modes, expanding the solution in terms of scalar and vector spherical harmonics. From a quantitative point of view, our findings reveal that the eyebulb vibration frequencies significantly depend on IOP. This dependency has two origins: "geometric" stiffening, due to an increase of the pre-stress, and "material" stiffening, due to the nonlinearity of the stress-strain curve of the sclera. The model shows that the second effect is by far dominant. We also find that the oscillation frequencies depend on ocular rigidity, but this dependency is important only at relatively large values of IOP. Thus close to physiological conditions, IOP is the main determinant of ocular vibration frequencies. The vitreous rheological properties are found to mostly influence vibration damping. This study contributes to the understanding of the mechanical behavior of the eye under dynamic conditions and thus has implications for non-contact intraocular pressure measurement techniques, such as acoustic tonometry. The model can also be relevant for other ocular pathological conditions, such as traumatic retinal detachment, which are believed to be influenced by the dynamic behavior of the eye.

## Introduction

The human eye is the organ that allows us to perceive visual information of the world around us. The shell enclosing the ocular bulb consists of three layers with different functions. The most external one is the fibrous corneo-scleral shell, which supports the mechanical loads acting on the organ. The vascular middle layer is the uvea, which consists of the iris, the ciliary body, and the choroid. Finally, the innermost layer is the retina, which is the nervous tunic, where the photoreceptors are located. The interior of the eye consists of three chambers: the anterior chamber, the posterior chamber and the vitreous chamber (Fig 1). The first is located between the cornea and the iris, the second is between the iris and the lens, and the third is delimited anteriorly by the lens and posteriorly by the retina. The anterior and posterior

Research (MIUR) through project PRIN 2022NNTZNM DISCOVER, and the "Departments of Excellence" initiative. Support from Istituto Nazionale di Alta Matematica – Gruppo Nazionale per la Fisica Matematica (INdAM-GNFM) is also acknowledged.

**Competing interests:** The authors have declared that no competing interests exist.

chambers are connected through the pupil and contain aqueous humor, a liquid with properties very similar to water. The vitreous chamber contains a gel, the vitreous humor, with visco-elastic properties [1].

The eye is a pressurized organ and the intraocular pressure (IOP) is responsible for the main mechanical loads acting on the corneo-scleral shell under physiological conditions. IOP is regulated by a delicate balance between the rate of aqueous production by the ciliary body [2] and resistance to its drainage at the junction between the cornea and the iris, mostly through a spongy tissue named trabecular meshwork [3]. IOP has a significant impact on the functioning of the eye and is involved in the onset and development of various pathological conditions. Most notably, elevated IOP increases the risk of developing glaucoma, a collection of eye conditions that can cause damage to the optic nerve and can result in vision loss [4].

Standard methods to measure IOP (contact tonometry methods) exploit the principle that the force required to deform the cornea by applanation or indentation increases with increasing IOP. The gold standard instrument in contact tonometry is the Goldmann applanation tonometer, which uses a probe to flatten a portion of the cornea and infers the IOP from the required force. Non-contact tonometers are also presently in use, which employ an air puff to flatten the cornea [5, 6]. A detailed review of the techniques presently in use to measure IOP and the underlying physical principles is reported in [7].

A significant limitation of standard tonometry methods is their inadequacy for self-administration. These traditional measurement techniques require professional operation and

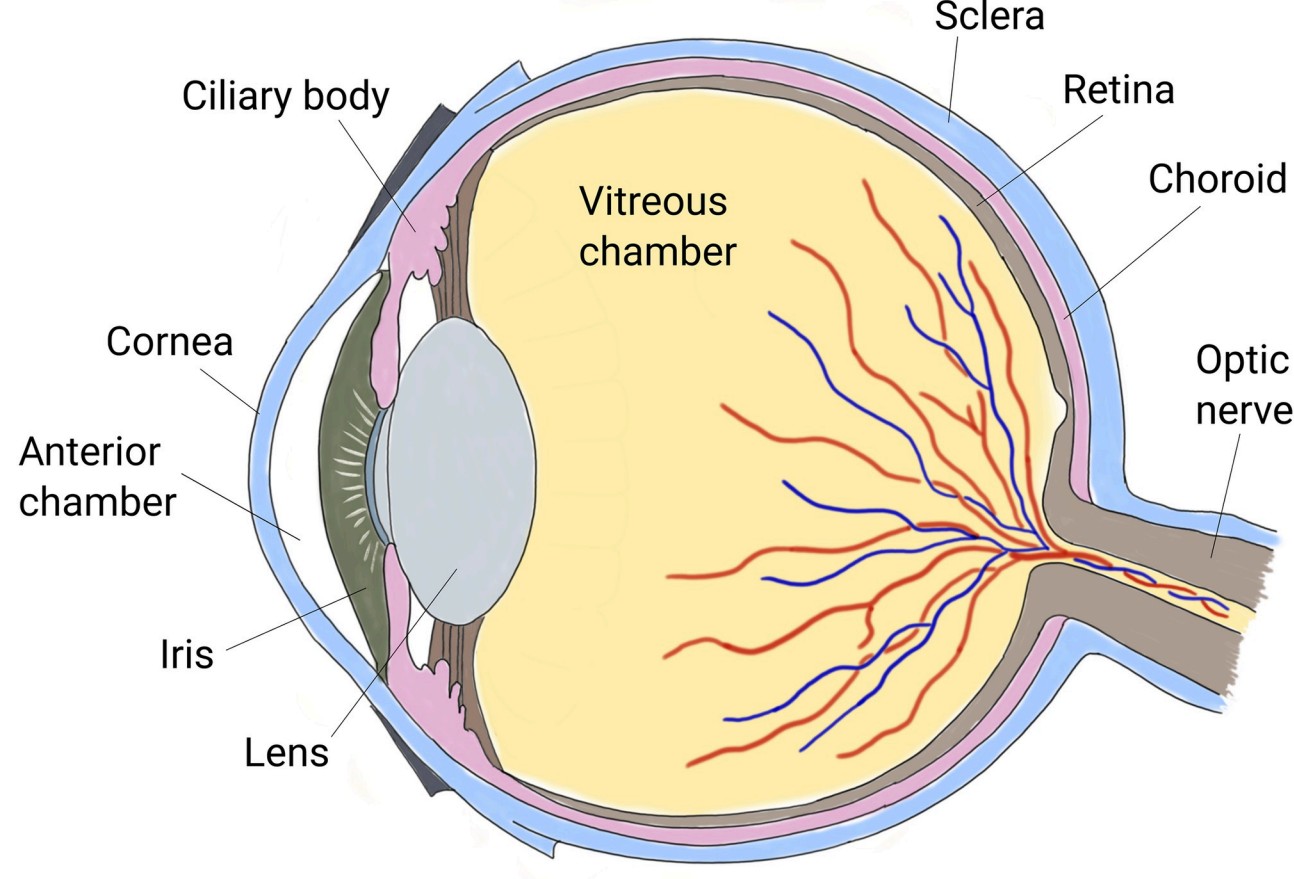

**Fig 1. Sketch of a cross-section of the human eye.**

precise application, making them unsuitable for individuals to use independently. A promising, non-contact IOP measurement technique, which has been investigated since the late seventies [8], and which could potentially be used to realize measuring devices that can be used autonomously by the patient, is acoustic tonometry. The general idea is to excite vibrations of the eye bulb with acoustic waves and measure its response, from which it is possible to infer the natural vibration frequencies of the eye and their damping ratio. These, in turn, are affected by IOP [8–13]. Although the preliminary results obtained adopting these techniques suggest that they have great potential, none of such approaches has made its way to the clinical practice yet.

The reliability of acoustic methods would certainly benefit from a better understanding of the mechanical behavior of the eye under dynamic conditions. The natural frequencies of the eye and their damping do not only depend on IOP but also on a number of other factors, such as mechanical properties and thickness of the corneoscleral shell, elasticity and viscosity of the vitreous body, size of the eye bulb, etc. Being able to quantify the role of each of these effects on the vibration frequencies is of key importance to isolate the role of IOP, which is something that a sound mathematical model can help do.

A better understanding of the dynamic response of the eye may also have an impact in other areas related to pathological conditions of the eye. For instance, retinal detachment is often a consequence of trauma or of vitreo-retinal tractions, which are both, to some extent, influenced by the dynamic behavior of the eye bulb and of the vitreous body [14–17]. It has been suggested that, due to its viscoelastic behavior, the vitreous body might serve as a mechanical damper for the eye, thus absorbing impacts, and protecting the lens and retina against mechanical injury [1]. Understanding the role of vitreous mechanical properties on ocular bulb dynamics provides useful information on how vitreous aging and vitreous replacement with tamponade substitutes would impact on tractions on the retina.

There is extensive literature on the mathematical modeling of spherical shell vibrations, which dates back to some classical works of the nineteenth century [18, 19]. The effect of a fluid filling an elastic sphere on its vibration frequencies was first considered by [20], and various further papers have been published since. A recent contribution that also summarizes previous results is [21]. The authors studied theoretically small oscillations of a pressurized, elastic, spherical shell subject to internal and external fluid effects.

Some authors have also studied theoretically or numerically the vibrations of a fluid-filled shell, specifically considering the problem of ocular bulb vibrations. In [22], the eye was modeled as an elastic shell, representing corneo-scleral shell, described with a realistic geometry. The shell was filled with an inviscid and incompressible fluid, representing the vitreous humor. The resulting model was solved using the finite-element method to compute the vibration modes of the eye and the dependence of resonant frequencies on IOP.

Salimi et al. [23] computed the natural frequencies of the eye using the finite-element method. They first described the eye as a spherical shell containing a fluid and then proposed an anatomically more accurate model. They also validated their numerical predictions against results from experimental tests.

Aloy at al. [24] proposed various models of the eye with increasing complexity and computed the oscillation frequencies of the system, with the aim of estimating indirectly the mechanical properties of ocular tissues. They first modeled the eye globe as a homogeneous sphere, then they accounted for the presence of an outer stiffer layer (the corneo-scleral shell) and, finally, modeled the cornea and sclera as distinguished tissues, with different mechanical properties.

Shih and Guo [25] also studied the natural modes of oscillation of the ocular bulb, described as a spherical elastic shell filled with an inviscid fluid. The theoretical model proposed in [25]

is obtained by adapting the equations that govern the equilibrium of a pre-stretched plate to a spherical geometry.

In this paper, we study coupled vibrations of the vitreous humor and corneo-scleral shell, modeling the former as a linear, viscoelastic, incompressible material, and the latter as a thin elastic spherical shell. In particular, the problem we want to solve is the following: given a set of material parameters available from experimental data, determine the dependence on IOP of the resonant frequencies of the least damped modes. In spite of the idealizations it is based on, our approach improves over the previous works in various respects. We account for the effect of pre-stress of the shell in a formally correct manner and this leads to governing equations for the shell that are slightly different from those derived in previous works.

We also consider the viscoelasticity of the vitreous body, which was invariably neglected in all previous contributions (in most cases the vitreous body was simply described as an inviscid fluid). This is likely to have an important effect, particularly on the damping properties of the system.

With these ingredients, we compute the resonant frequencies and the vibration modes of the eyebulb, highlighting the role of IOP, stiffening of the sclera, and damping associated with the viscoelastic behavior of the vitreous. This allows us to assess the importance of pressure and stiffening on the resonant frequencies, as well as the effect of the rheological properties of the enclosed fluid on the damping rate. The importance of these effects is discussed in the final section of this paper, where we summarize our main findings.

## Materials and methods

### Formulation of the mathematical model

To compute the vibration properties of the eyebulb we have developed an analytical model, where the eyebulb is described as an elastic pre-stressed, spherical shell (the corneo-scleral shell) filled with an incompressible viscoelastic material (the vitreous humor).

The equations that govern the motion in the interior of the eyebulb are the standard ones for linear viscoelasticity, (1). They have been used in [15] to characterize the vibrations of the vitreous body, under the assumption that the cornea was rigid.

For the shell, we use the coordinate-free approach developed in [26] and we adopt the equations of motion for a pressurized spherical shell, (2), developed therein. We refer to [26] and to the S1 File for additional information concerning the shell model.

The two models (vitreous and corneo-scleral shell) are coupled using the no-slip condition and also assuming that the shell is loaded by the traction locally exerted by the inner viscoelastic material. More in detail, we consider an equilibrium state where the eyebulb is at rest, with an internal constant pressure $p$. The corneoscleral shell is thus in a stressed state, described by membrane force-tensor $\mathring{\mathbf{N}} = \frac{pR}{2}\mathbf{P}$, where $\mathbf{P} = \boldsymbol{I} - \boldsymbol{n} \otimes \boldsymbol{n}$, with $\boldsymbol{I}$ the identity tensor and $n$ the outward unit normal, is the projector on the tangent plane to the shell. Note that in this state the bending moment vanishes.

Small-amplitude vibrations are described by a displacement field $\boldsymbol{u}(\boldsymbol{x}, t)$, with $\boldsymbol{x}$ denoting the position vector and $t$ time. In the vitreous humor, the displacement obeys the motion equations

$$\rho_{\mathrm{v}}\ddot{\boldsymbol{u}} = \mathrm{div}\boldsymbol{S},$$

$$\mathrm{div}\boldsymbol{u} = \boldsymbol{0},$$

(1)

where each superimposed dot represents partial differentiation with respect to time, $\rho_{\mathrm{v}}$ is the density of the vitreous humor and $\boldsymbol{S}$ is the *increment of the nominal (Piola) stress*. On the

corneoscleral shell, the normal and tangential components of the displacement, respectively, $w = \boldsymbol{n} \cdot \boldsymbol{u}$ and $\mathbf{v} = \mathbf{P}\boldsymbol{u}$, obey the motion equations

$$\rho_s h \ddot{w} = \operatorname{div}_s(\mathbf{P} \operatorname{div}_s \mathbf{M}) - \frac{1}{R}\mathbf{P} \cdot \mathbf{N} + \frac{p}{2}R\Delta_s w - \frac{p}{2}\operatorname{div}_s \mathbf{v} - \boldsymbol{n} \cdot \boldsymbol{Sn},$$

$$\rho_s h \ddot{\mathbf{v}} = \mathbf{P} \operatorname{div}_s \mathbf{N} + \frac{1}{R}\mathbf{P}(\operatorname{div}_s \mathbf{M}) - \mathbf{P}\boldsymbol{Sn}, \tag{2}$$

where $\mathbf{N}$ and $\mathbf{M}$ are the increments of the *nominal membrane-force tensor* and *bending-moment tensor*. In the above equations, $\rho_s$ denotes the shell density, $h$ the shell wall thickness, $R$ the reference shell radius, $p$ the internal reference pressure (IOP) and $\operatorname{div}_s$ the divergence operator on the surface of the shell.

On the right-hand sides of Eq (2), the last terms represent the force per unit reference area exerted by the vitreous humor on the corneoscleral shell. Moreover, the first terms on the right-hand sides of the same equation set represent the extra contribution due to the shell pre-tension due to the IOP.

Within the vitreous, we adopt the following constitutive equation for the incremental nominal stress:

$$\boldsymbol{S} = p\nabla\boldsymbol{u}^{\top} + \boldsymbol{\Sigma}, \tag{3}$$

where

$$\boldsymbol{\Sigma} = -q\boldsymbol{I} + 2\int_{-\infty}^{t} G(t-s)\boldsymbol{D}(s)\mathrm{d}s, \tag{4}$$

with $q$ the pressure increment, $\boldsymbol{D} = \operatorname{dev}(\operatorname{sym}\nabla\dot{\boldsymbol{u}})$ the strain rate and $G(t)$ the stress relaxation function. The first term on the right-hand side of (3) accounts for the pre-compression associated with the IOP. An explanation of the nature of this term in connection with the definition of nominal stress may be found in [27].

The stress relaxation function $G(t)$ embodies the information of the material that fills the cavity, in response to a stress relaxation test. For an elastic material, $G(t)$ is proportional to the Heaviside function; for a viscous fluid, $G(t)$ is proportional to the Dirac function, centered at the origin [28]. In the case of harmonic motion, the properties of the stress relaxation function are encoded in the complex modulus

$$\mathsf{G}(\zeta) = \zeta \int_0^{\infty} e^{-\zeta\tau}G(\tau)\mathrm{d}\tau. \tag{5}$$

We assume that the stress-response function of the vitreous can be described with the Kelvin-Voigt model (a linear spring and a dashpot arranged in parallel), so that

$$\mathsf{G}(\zeta) = \gamma + \zeta\eta, \tag{6}$$

where $\gamma$ is the shear modulus and $\eta$ is the viscosity.

For the corneoscleral shell we adopt the constitutive equations

$$\mathbf{N} = \frac{pR}{2}\nabla\boldsymbol{u} + h(2\mu\boldsymbol{\varepsilon} + \tilde{\lambda}(\operatorname{tr}\boldsymbol{\varepsilon})\mathbf{P}) + \frac{1}{R}\mathbf{M},$$

$$\mathbf{M} = \frac{h^3}{12}(2\mu\boldsymbol{\kappa} + \tilde{\lambda}(\operatorname{tr}\boldsymbol{\kappa})\mathbf{P}), \tag{7}$$

where $\mu$ and $\tilde{\lambda}$ are material moduli, which are linked to the Lamé constants and to Young's

modulus and Poisson's ratio through eqautions (9). The constant $\tilde{\lambda}$ is a correction to one of the Lamé parameters that accounts for the fact that a thin shell is in a plane stress state. Moreover,

$$\boldsymbol{\varepsilon} = \frac{1}{2}(\mathbf{P}\nabla_s\mathbf{v} + \nabla_s\mathbf{v}^\top\mathbf{P}) + \frac{w}{R}\mathbf{P},$$

$$\boldsymbol{\kappa} = -\mathbf{P}\nabla_s\nabla_s w + \frac{1}{R}\mathbf{P}\nabla_s\mathbf{v} + \frac{1}{R}\nabla_s\mathbf{v}^\top\mathbf{P} + \frac{w}{R^2}\mathbf{P}, \tag{8}$$

are, respectively, the in-plane stretch and bending tensors and $\nabla_s$ is the surface gradient operator. Eqs (2), (7) and (8) follow from a systematic linearization of the equations that govern the dynamics of non-linear elastic shells [26]. The first term on the right-hand side of $(7)_1$, as well as the two pressure-dependent terms on the right-hand side of (2), yield an extra restoring term in the equations of motion, which results in the change of the effective stiffness. We shall demonstrate that such a change is partially responsible for the shift of resonant frequency that accompanies the increase of intraocular pressure.

The equations that govern the dynamics of an empty or a fluid-filled spherical shell have been derived and studied by several authors. Lamb [19] studied the small-amplitude vibrations of a thin spherical shell by fully solving the dynamical equations of elasticity in a domain bounded between two concentric spherical surfaces. The vibrations of an elastic spherical shell containing a fluid have been studied first by Love [29], then by Rand and DiMaggio [20] by Engin and Liu [30], and by Bai and Wu [31]. The effect of viscosity of the enclosed fluid has been investigated by Su [32].

The explicit contribution of an initial pressure to the motion of the shell has been considered by Kuo et al. [21] and by Shih and Guo [33]. In both cases, the initial pressure results in an extra term that adds up to the restoring elastic forces. Kuo et al. [21] take as starting point the equations that govern the axisymmetric motions of a spherical shell and introduce the restoring force due to the pre-stress by an insightful ad-hoc argument. Shih and Guo [33], instead, take as starting point the equations of a pre-stressed membrane taken from [34], and take into account the effect of curvature replacing the Laplacian operator of the membrane with the Laplace-Beltrami operator. Compared with these two references, our approach to the calculation of the extra restoring force due to the IOP is based on a systematic linearization of the nonlinear equation of motions derived in [26], and yields slightly different equations. In particular, in our case the initial pre-stress affects also the tangential motion of the shell. However, the extra contribution of the normal component of the displacement that appears in our motion equations (see the S1 File), proportional to the surface Laplacian $\Delta_s$ of the normal displacement, coincides with that of Kuo et al. [21] and that of Shih and Guo [33].

## Parameter values

All parameter values that have been used in the model are reported in Table 1.

Jesus et al. [35] measured the scleral radius through an approximation of the topographical scleral data to a sphere and found the value of 11.2±0.3 mm, which is what we use in the model.

The thickness of the sclera is highly variable from point to point, ranging from 0.50 mm at the limbus to 0.95 mm at the posterior pole (see Table 1). The thickness of the central cornea is approximately 0.56 mm. In our model, the corneo-scleral shell is modeled as a constant thickness structure, and we have adopted the value 0.5 mm, which is in line with the value chosen in related studies [25].

**Table 1. Experimental values of the parameters.**

| Parameter | Value | Reference |
|---|---|---|
| Radius of curvature of the sclera | 11.2 ± 0.3 | [33] |
| Thickness of the sclera | 0.50 ± 0.11 mm limbus | [36] |
| | 0.43 ± 0.14 mm ora serrata | |
| | 0.42 ± 0.15 mm equator | |
| | 0.65 ± 0.15 mm posterior region | |
| | 0.95 ± 0.18 mm posterior pole | |
| | 0.86 ± 0.21 mm optic nerve region | |
| Thickness of the cornea | 0.561 ±0.026 mm | [37] |
| Density of the sclera | 1077 ± 5 kg/m$^3$ | [38] |
| Density of the cornea | 1058 ± 7 kg/m$^3$ | [38] |
| Shear modulus of the vitreous | 10 Pa | [15, 39] |
| Viscosity of the vitreous $\eta$ | 0.39 Pa · s | [15, 39] |
| Ocular rigidity | 0.021 $\mu$l$^{-1}$ | [40, 41] |
| Physiological IOP | 15 mmHg | [42] |

Densities of the sclera and cornea have been taken equal to 1077 and 1058 kg/m$^3$, respectively [38].

The viscoelastic properties of the vitreous have been measured by several authors; see Table 1 in [1]. Most authors have characterized vitreous properties through the complex modulus G, defined by Eq (5). In this work we interpret the rheological tests using the Kelvin-Voigt model (6) and adopt the values of $\gamma$ = 10 Pa and $\eta$ = 0.39 Pa · s, that are derived from measurements by [39] (see Table 1 in [15]).

We finally need to specify the values of the parameters $\mu$ and $\tilde{\lambda}$. In conventional engineering theories, these parameters are the shear modulus and the effective first Lamé constant. They are given by

$$\mu = \frac{E}{2(1+v)} \quad \text{and} \quad \tilde{\lambda} = \frac{2Ev}{1+v}, \tag{9}$$

where $E$ and $v$ are, respectively, the Young's modulus and the Poisson's ratio and $\tilde{\lambda}$ represents a correction to the Lamé parameter $\lambda = \frac{Ev}{(1+v)(1-2v)}$. If the material is incompressible, then $v$ = 0.5 and (9) become $\mu = \frac{1}{3}E$ and $\tilde{\lambda} = \frac{2}{3}E$, so that

$$\tilde{\lambda} = 2\mu. \tag{10}$$

It would be tempting to employ (9) or (10) to fit our parameters, employing the available measurements of the Young's modulus and Poisson's ratio of the cornea. However, the available data are scattered and the outer shell of the eye is highly anisotropic and non-homogeneous. In addition, one should consider that the mechanical and geometrical properties of the cornea are significantly different from those of the sclera [43, 44].

For the aforementioned reasons, we have opted for fitting the parameters $\mu$ and $\tilde{\lambda}$ using measurements from inflation tests that provide a global estimate of the bulb mechanical properties, somehow averaging over the spatial variability of tissue properties. Friedenwald [40]

proposed the following empirical law to link the ocular volume $V$ to IOP $p$

$$\log\left(\frac{p}{p_0}\right) = K(V - V_0),\tag{11}$$

where $p_0$ and $V_0$ are the corresponding reference values, and $K$ is a constant called *ocular rigidity*. Friedenwald [40] estimated the ocular rigidity to be 0.021 $\mu\text{l}^{-1}$.

It follows from (11) that $\frac{dp}{dV} = Kp$, and hence

$$\frac{dp}{dR} = 4\pi R^2 Kp.\tag{12}$$

For a sphere of radius $R$, a uniform increment $dR$ of the radius corresponds to a normal displacement $w = dR$ and to a tangential displacement $\mathbf{v} = \mathbf{0}$. Thus, by (8), the stretching and bending strains are, respectively,

$$\boldsymbol{\varepsilon} = \frac{dR}{R}\mathbf{P}, \qquad \boldsymbol{\kappa} = \frac{dR}{R^2}\mathbf{P}.\tag{13}$$

Neglecting bending moments, the increment of the nominal membrane force tensor is, according to the constitutive Eq (7),

$$\mathbf{N} = \left(\frac{pR}{2} + 2h(\mu + \tilde{\lambda})\right)\frac{dR}{R}\mathbf{P}.\tag{14}$$

The corresponding increment of nominal traction (force per reference unit area) is $\mathbf{b} = \left(dp + p\frac{dR}{R}\right)n$, thus, the equilibrium equation in the normal direction (see (2)) yields

$$-\left(p + 4\frac{h}{R}(\mu + \tilde{\lambda})\right)\frac{dR}{R} + dp + p\frac{dR}{R} = 0,\tag{15}$$

whence

$$\frac{dp}{dR} = 4\frac{h}{R^2}(\mu + \tilde{\lambda}),\tag{16}$$

which corresponds to

$$\frac{dV}{dp} = \frac{\pi R^4}{h(\mu + \tilde{\lambda})} = C,\tag{17}$$

where $C$ is ocular compliance. Comparison of (12) and (16) yields

$$\mu + \tilde{\lambda} = \pi\frac{R^4}{h}Kp, \qquad C = \frac{1}{Kp}.\tag{18}$$

Since the sclera is an almost incompressible material, we assume that the Poisson coefficients be equal to 0.5. Then, (10) and (18) yield

$$\mu = \frac{\pi}{3}R^3 Kp.\tag{19}$$

As a consequence, the value of the Young's modulus that results from (9) and (19) is given by

$$E = \pi\frac{R^4}{h}Kp,\tag{20}$$

in agreement with [45]. In particular, taking $R$ = 11.2 mm, $h$ = 0.5 mm, $K$ = 0.021 $\mu$l$^{-1}$ and $p$ = 15 mmHg, we obtain $E$ = 4.15 MPa. This is in line with experimentally determined values, see Table 1 in [43].

We note that the above expressions imply that the Young's modulus $E$ increases (and the ocular compliance $C$ decreases) as IOP grows. Thus, by assuming a constant value of ocular rigidity $K$, we effectively account for corneo-scleral tissue stiffening in response to tissue strain.

## Solution procedure

The manipulations needed to find a solution of the mathematical model described in the previous section are quite elaborated, and a detailed description is reported in the S1 File. Here we just outline the main steps in the following.

The first step of the solution process consists in passing from the time domain to the frequency domain, by writing

$$\boldsymbol{u}(x,t) = \mathrm{Re}(e^{\zeta t}\mathsf{u}(x,\zeta)), \quad q(x,t) = \mathrm{Re}(e^{\zeta t}\mathsf{q}(x,\zeta)), \tag{21}$$

where $\zeta \in \mathbb{C} \setminus \{0\}$ is a complex frequency. The imaginary part of $\zeta$ is the angular frequency of oscillation of the solution, while the opposite of the real part yields the rate of decay of the solution.

We expand pressure increment and displacement using scalar and vector spherical harmonics, respectively (for the detailed definitions we refer to the S1 File and [46, 47]). Specifically, we write

$$\mathsf{q}(\boldsymbol{r}) = \sum_{\ell \geq 0} \sum_{-\ell \leq m \leq \ell} Q_{\ell m}(r,\zeta) Y_{\ell m}(\hat{\boldsymbol{r}}), \tag{22}$$

and

$$\mathsf{u}(\boldsymbol{r},\zeta) = \sum_{\ell \geq 0} \sum_{-\ell \leq m \leq \ell} \mathbf{u}_{\ell m}(\boldsymbol{r},\zeta), \tag{23}$$

where

$$\mathsf{u}_{\ell m}(\boldsymbol{r},\zeta) = P_{\ell m}(r,\zeta)\mathbf{p}_{\ell m}(\hat{\boldsymbol{r}}) + B_{\ell m}(r,\zeta)\mathbf{b}_{\ell m}(\hat{\boldsymbol{r}}) + C_{\ell m}(r,\zeta)\mathbf{c}_{\ell m}(\hat{\boldsymbol{r}}). \tag{24}$$

Here $\boldsymbol{r}$ represents the position with respect to the center of the sphere; $r = |\boldsymbol{r}|$ is the distance from the center, and $\hat{\boldsymbol{r}}$ is the unit vector pointing in the direction of $\boldsymbol{r}$. The functions $Y_{\ell m}$ are the spherical harmonics, while $\mathbf{p}_{\ell m}$, $\mathbf{b}_{\ell m}$ and $\mathbf{c}_{\ell m}$ are the vector spherical harmonics, defined on the unit sphere. We recall that $\mathbf{p}_{\ell m}$ are radial vectors and $\mathbf{b}_{\ell m}$ and $\mathbf{c}_{\ell m}$ are vectors tangential to the sphere surface and orthogonal to each other.

As shown in the S1 File, the substitution of (21)-(24) into the motion Eqs (1), (3) and (4) yields a system of ordinary differential equations for the coefficients $Q_{\ell m}$, $P_{\ell m}$, $B_{\ell m}$ and $C_{\ell m}$. For $\ell = 0$, this system admits only the trivial solution. For $\ell \geq 1$, bounded solutions have the

general form:

$$
\begin{aligned}
Q_{\ell m}(r,) &= -C_{\ell m}^{(1)} \frac{\rho_v R \zeta^2}{\ell} \left(\frac{r}{R}\right)^{\ell}, \\[2mm]
P_{\ell m}(r,) &= C_{\ell m}^{(1)}\left(\frac{r}{R}\right)^{\ell-1} + C_{\ell m}^{(2)}\left(\frac{r}{R}\right)^{-1} j_{\ell}\left(a(\zeta)\frac{r}{R}\right), \\[2mm]
B_{\ell m}(r,) &= C_{\ell m}^{(1)} \frac{s_{\ell}}{\ell}\left(\frac{r}{R}\right)^{\ell-1} + \frac{C_{\ell m}^{(2)}}{s_{\ell}}\left(a(\zeta)\, j_{\ell-1}\left(a(\zeta)\frac{r}{R}\right) - \ell\left(\frac{r}{R}\right)^{-1} j_{\ell}\left(a(\zeta)\frac{r}{R}\right)\right), \\[2mm]
C_{\ell m}(r,) &= C_{\ell m}^{(3)} j_{\ell}\left(a(\zeta)\frac{r}{R}\right),
\end{aligned}
\tag{25}
$$

where $C_{\ell m}^{(1)}$, $C_{\ell m}^{(2)}$ and $C_{\ell m}^{(3)}$ are three arbitraty constants, $a(\zeta) = iR\zeta\sqrt{\rho_v/\mathsf{G}(\zeta)}$, and $j_{\ell}$ is the $\ell$-th spherical Bessel function, defined by $j_{\ell}(x) = \sqrt{\pi/(2x)}\,\mathsf{J}_{\ell+1/2}(x)$.

Upon substitution of the solution (25) into the motion equations (2) that govern the dynamics of the corneo-scleral shell we obtain two characteristic equations, namely,

$$
\begin{pmatrix} m_{11}(\ell,\zeta) & m_{12}(\ell,\zeta) \\ m_{21}(\ell,\zeta) & m_{22}(\ell,\zeta) \end{pmatrix} \begin{pmatrix} C_{\ell m}^{(1)} \\ C_{\ell m}^{(2)} \end{pmatrix} = \mathbf{0},
\tag{26}
$$

and

$$
m_{33}(\ell,z) C_{\ell m}^{(3)} = 0,
\tag{27}
$$

where $m_{ij}(\ell,\zeta)$ are complex coefficients, the expressions of which are reported in the S1 File. Vibration frequencies are determined by imposing either that the determinant of the matrix in (26) vanishes, that is,

$$
m_{11}(\ell,\zeta) m_{22}(\ell,\zeta) - m_{21}(\ell,\zeta) m_{12}(\ell,\zeta) = 0,
\tag{28}
$$

or

$$
m_{33}(\ell,\zeta) = 0.
\tag{29}
$$

The characteristic equations (28) and (29) are nonlinear in the eigenvalues $\zeta$ and are solved numerically using the MATLAB (MathWorks®) function `fsolve`. This function uses Powell's dog leg algorithm [48]. Solutions of these equations define two families of vibration modes. Modes in the first family are a linear combination of the harmonics $\mathbf{p}_{\ell m}$ and $\mathbf{b}_{\ell m}$, through the functions $P_{\ell m}$ and $B_{\ell m}$, which depends on the coefficients $C_{\ell m}^{(1)}$ and $C_{\ell m}^{(2)}$. These coefficients are obtained by solving (26). For this class of modes, the displacement field on the shell has both normal and tangential components.

The second family involves only the vector spherical harmonics $\mathbf{c}_{\ell m}$, and the corresponding velocity field is tangential to the shell surface.

For $\ell = 1$, modes in the first family are singular at the origin [15, 47], and hence must be discarded. Still for $\ell = 1$, vibration modes in the second family have the special property that every concentric sphere within the ball undergoes a rigid oscillatory rotation. Such modes have been already studied in [15], and do not entail any deformation of the shell. As such, they are not detectable by any method based on measuring the deformation of the corneoscleral shell, and hence they are not of interest in the context of the present investigation.

We remark that the index $m$ does not appear in the characteristic equations. This is because harmonics having the same $\ell$, but different $m$, can be transformed into each other through a

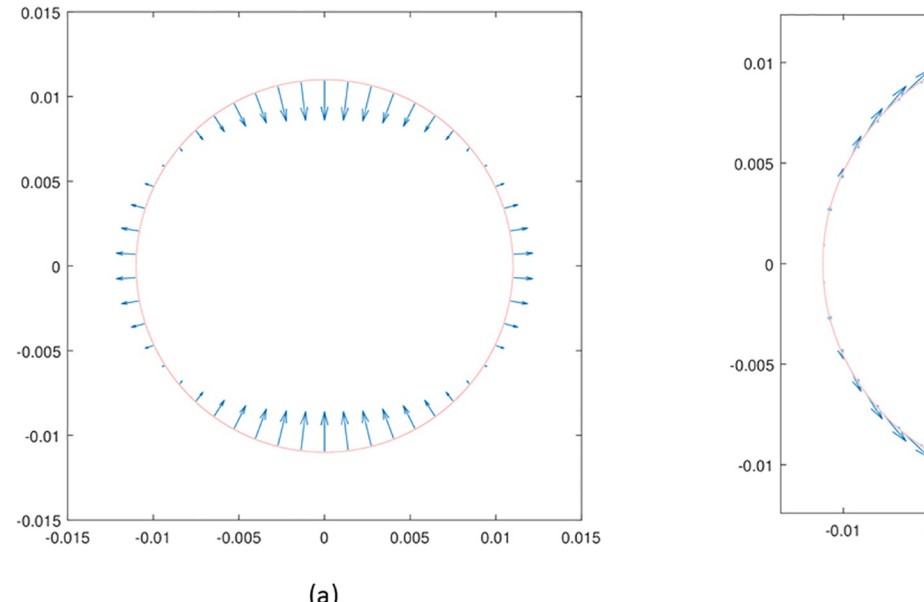
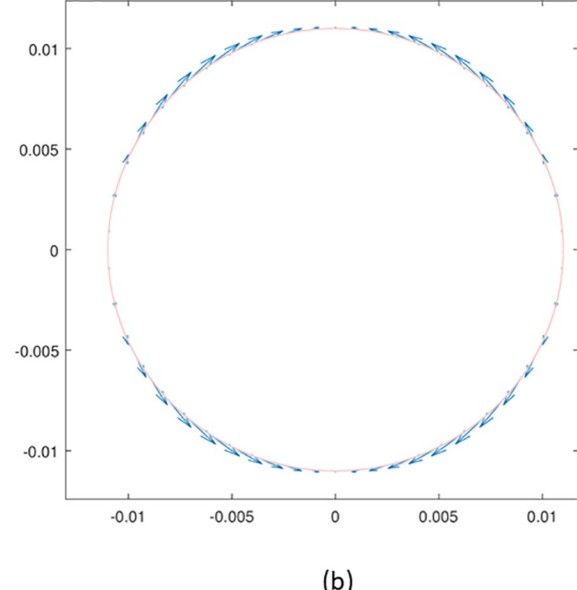

(a)                                                    (b)

**Fig 2. Velocity fields on the boundary corresponding to the two eigenvalues from (28) for $\ell = 2$ and $m = 0$.**

rotation, which makes them physically equivalent [15]. For this reason, we focus our attention on the case $m = 0$.

In Fig 2 we represent the velocity of the two lowest-frequency modes ($\ell = 2$) in the first family. The first mode involves mainly bending of the shell, whereas the second involves stretching since the velocity field is almost tangential. In the figures we show only a meridian section of the sphere as the motion is axisymmetric.

Fig 3 shows is a three-dimensional representation of the vibration mode for $\ell = 2$ of the second family. This mode of oscillation is purely tangential and involves a twist deformation of the shell.

## Validation of the model

In [23], Salimi et al. studied free vibrations of the eyeball using a FEM model. To calibrate their model, they performed experiments on the vibrations of a water-filled elastic ball. In this section, we validate our analytical model against these experimental measurements.

The ball was filled with water through an injector and the internal pressure was measured with a pressure gauge. The ball was suspended to an elastic cord so that it could freely vibrate. A little hammer was used to generate ball vibrations, which were measured using an accelerometer. The authors determined the vibration spectrum in response to the excitation input, from which they inferred the main natural oscillation frequencies of the sphere. They performed experiments for three different values of the internal water pressure.

A comparison between their experimental findings and the results predicted by our model is shown in Table 2, where we report the vibration frequencies of the lowest mode, with $\ell = 2$. In our computations, we used to the following parameters [23]: sphere radius 25 mm; thickness of the wall 4 mm; Young's modulus 4.8 MPa, Poisson coefficient 0.45; shell density 1200 kg/m$^3$; water density 1000 kg/m$^3$; water viscosity $10^{-3}$ Pa · s.

The error of the model is always lower than 10%. Both experiments and model predict an approximately linear dependency of the frequency from the pressure. Given some possible

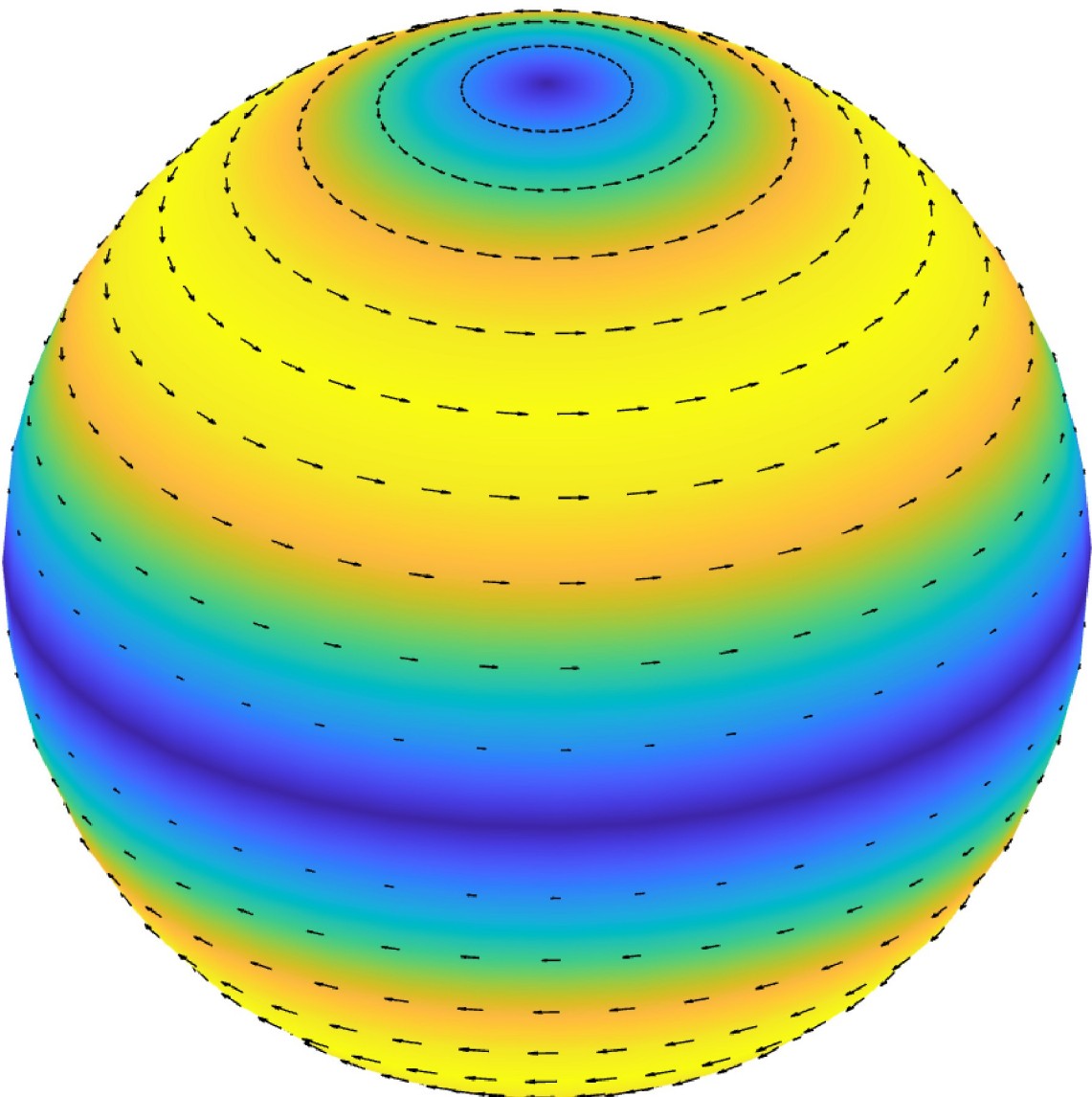

**Fig 3. Three-dimensional rendering of the velocity field on the boundary corresponding to the eigenvalue from (29) for ℓ = 2 and m = 0.**

**Table 2. Comparison between the model predictions and Salimi et al.'s [47] experiments on water-filled elastic ball.**

| Pressure | 13.7 kPa | 34.5 kPa | 62.0 kPa |
|---|---|---|---|
| **Experimental frequencies** | 187 Hz | 195 Hz | 206 Hz |
| **Computed frequencies** | 176 Hz | 180 Hz | 187 Hz |
| **Error %** | 5.88% | 8.33% | 9.22% |

experimental uncertainty regarding material properties of the sphere we regard this agreement acceptable. The increasing error with rising pressure can be attributed to the nonlinear behavior of rubber, leading to an increasing Young's modulus, compared to the reference value of 4.8 MPa used in the calculations, as the internal pressure grows.

## Results

We begin by showing the lowest frequency eigenvalues of the system, obtained by solving the characteristic equations (28) and (29), with $m = 0$ and $\ell = 2$. As discussed above, these eigenvalues are our main interest in this work. Results are reported in Fig 4 for various cases, relative to different materials filling the eyebulb. The vertical and horizontal axes represent the imaginary and the real part of the eigenvalue $\zeta$, respectively; these correspond, in the order, to the oscillation frequency and to the damping ratio. Empty circles represent the vibration frequencies of the empty shell; in this case, damping vanishes as the shell is assumed to have a purely elastic behavior and does not dissipate energy by itself. Mathematically, this implies that the corresponding eigenvalues are purely imaginary. The eigenvalue with the lowest frequency corresponds to the bending mode, shown in Fig 2a, obtained as one of the solutions of (28). The second eigenvalue (frequency of ≈1000 Hz) corresponds to the purely tangential mode, shown in Fig 3 and obtained by solving equation (29). The highest frequency eigenvalue corresponds to the third solution of (28), whose associated mode is shown in Fig 2b.

The presence of a fluid inside the shell increases the total mass of the system and, as can be seen in the figure, lowers considerably the vibration frequencies. Moreover, the system becomes dissipative when it contains a viscous material. This implies that all corresponding eigenvalues are now complex, with the real part being negative, which corresponds to dissipation. In all cases, the lowest frequency is associated with the mode corresponding to the first root of the characteristic equation of (28) (Fig 2a), which is thus the mode we will mostly focus our attention on in the following. This is because this mode is the one which will survive the longest time, after excitation of the eye bulb vibration.

The various points with different colors in Fig 4 correspond to cases of clinical interest, in which the eyeball is filled with healthy vitreous (blue), water (red) and silicone oil (yellow). The case of water is representative of a completely liquefied vitreous, and silicone oils are often

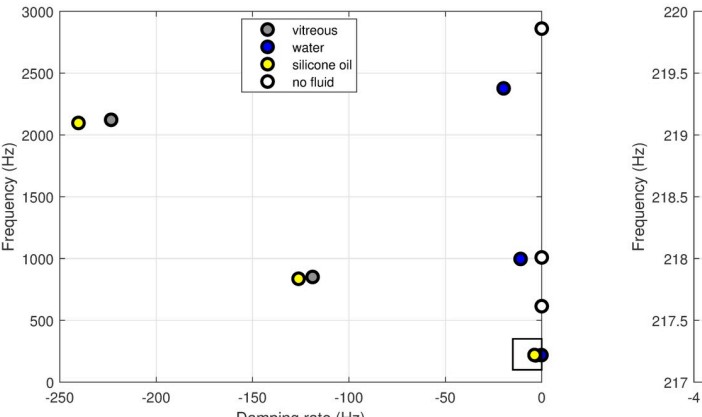 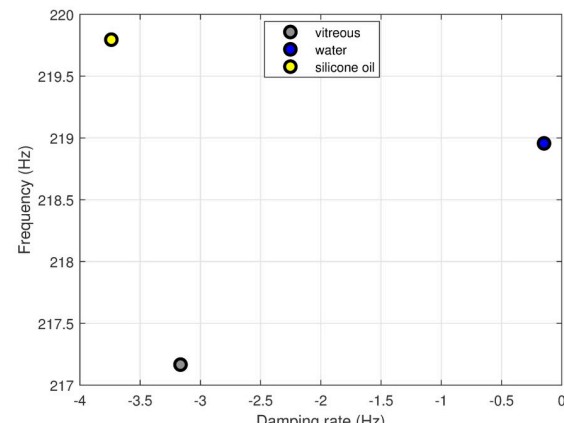

**Fig 4. Comparison between the complex frequencies for different materials filling the vitreous cavity.** Water: density of 1000 kg/m³ and viscosity of $10^{-3}$ Pa · s; vitreous body: values from Table 1 and density of 1000 kg/m³; silicone oil: density of 970 kg/m³ and viscosity of 0.5 Pa · s.

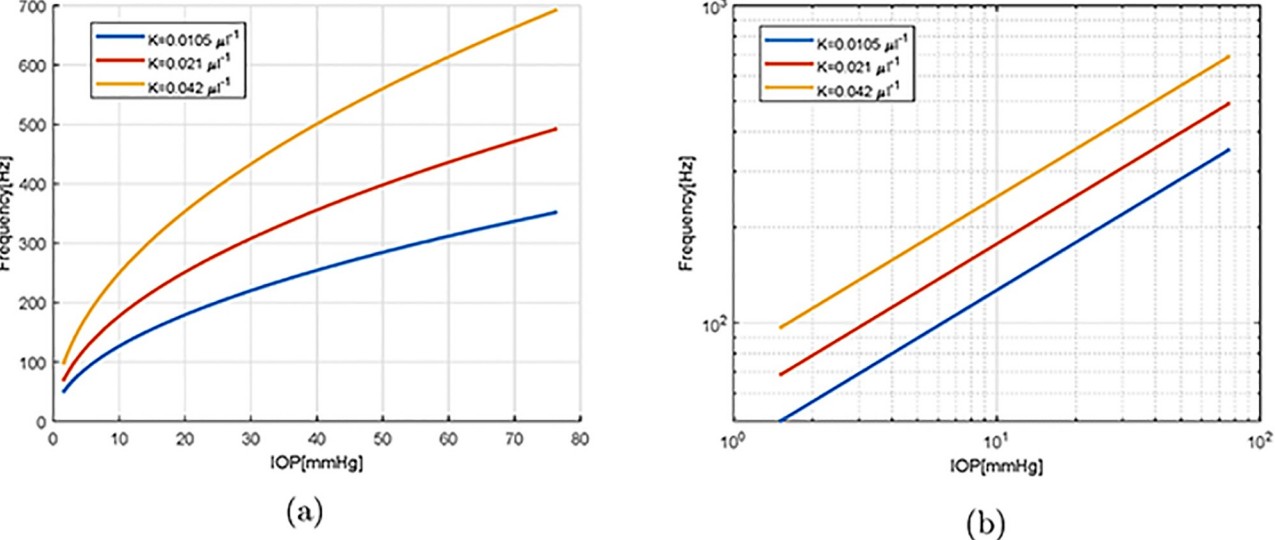

**Fig 5. Dependency of the frequency of oscillation of the slowest decaying mode as a function of IOP, for 3 different values of the ocular rigidity *K*.** (a) linear scales, (b) log-log scales.

used as vitreous replacement fluids after vitrectomy. The mechanical properties adopted for each case are reported in the caption of Fig 4. The figure shows that the frequencies of oscillation are weakly affected by the material filling eyebulb. On the contrary, the damping rate strongly depends on the viscosity of the filling material, being much higher for silicone oil and the real vitreous than for water.

In Fig 5 we investigate the dependency of the vibration frequency of the less damped mode on IOP, for different values of ocular rigidity *K*. The oscillation frequency increases markedly as IOP is raised from very small values up to 80 mmHg (which is an exceedingly high value). The dependency is particularly strong at low IOP. The results reported Fig 5 also show that ocular stiffness has a significant impact on the vibration frequencies, especially at large values of IOP. The intermediate curve, which corresponds to the value of ocular rigidity proposed by Friedenwald, is in good agreement the curve from Fig 5 of Ref. [8].

In Fig 5(b) we report the same curves as on Fig 5(a) but in a log-log plot. This shows that, on such a plane, the curves become straight lines, which means that the oscillation frequency depends on IOP according to a power law. Moreover, the curves are almost parallel to each other. In the case of the empty shell, an argument based on dimensional analysis shows that, under the assumption that the ocular rigidity *K* is constant, i.e. Young's modulus grows linearly with IOP according to (20), the oscillation frequency depends on the square root of IOP. This is confirmed by our solution. Interestingly, the angular coefficient of the curves in Fig 5(b) is very close to 1/2, which implies that the presence of vitreous within the sphere does not modify significantly this dependency.

We point out that, according to equation (20), Friedenwald's law (11) implies that the corneo-scleral tissue stiffens with stretch. To understand the importance of such an effect, we compare in Fig 6 the IOP-frequency curve for $K = 0.021 \ \mu l^{-1}$ from Fig 5 with the curve obtained by assuming a constant Young modulus (and thus also a constant ocular compliance *C*), determined by equation (20) for *p* = 15 mmHg. The comparison between the two curves shows that neglecting ocular stiffening results in a substantial underestimation of the change

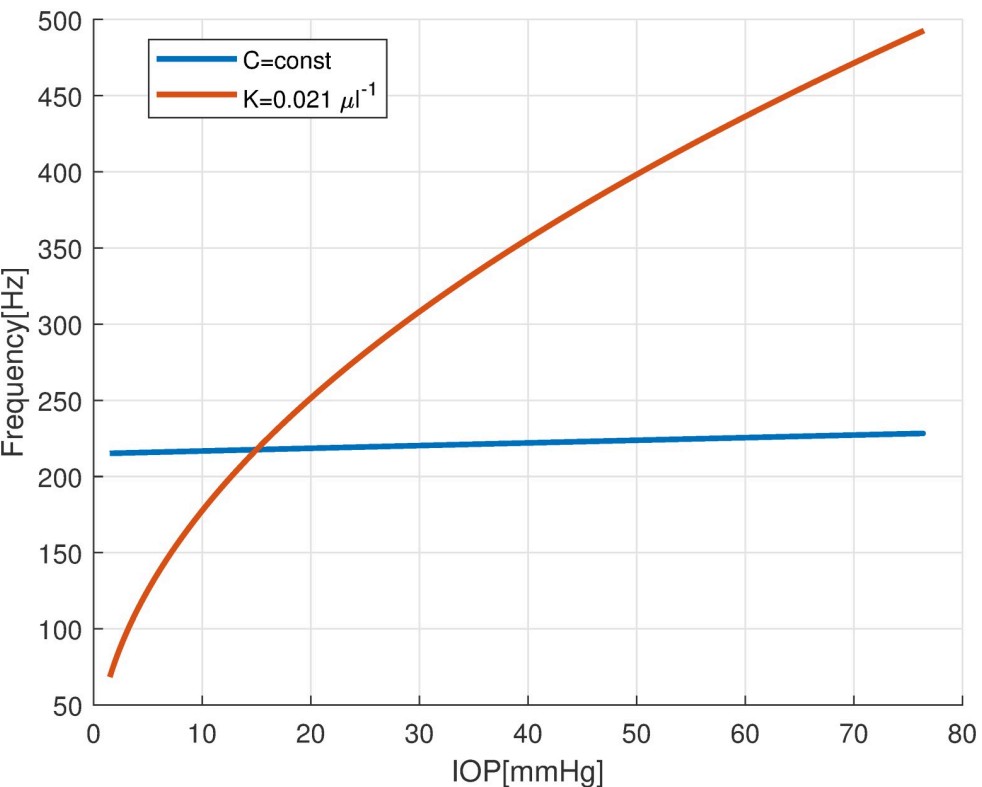

**Fig 6. Dependency of the frequency of oscillation of the slowest decaying mode as a function of IOP.** Comparison between the model with constant ocular compliance $C$ (blue) and the model with constant ocular rigidity $K$ (red).

of vibration frequency with IOP. This confirms the importance of tissue stiffening effects, which have been already pointed out by Alarm et al. [9].

The effect of viscosity on the vibration damping rate is shown in Fig 7, for the case in which the eyeball is filled with a viscous fluid. The viscosity in the plot is varied from $10^{-6}$ m$^2$/s, which is representative of water, to $10^{-3}$ m$^2$/s, which a typical viscosity of silicone oils used as tamponade fluids after vitrectomy.

Finally, in Fig 8 we plot contour lines of the frequency (a) and damping rate (b) of the slowest decaying mode as a function of IOP and ocular rigidity $K$. In the figure, IOP is varied over a very wide range of values and $K$ is modified by ±50% from the baseline value 0.021 $\mu$l$^{-1}$. The plots show that at low values of IOP, ocular rigidity has little effect on the eigenvalues of the system as the contour lines are almost vertical. As IOP increases, ocular rigidity becomes progressively more relevant.

## Discussion and conclusions

We have developed a mechanical model of the vibrations of the eyebulb, which describes the cornea and the sclera as a thin elastic shell with pre-stress, and the vitreous humor as a visco-elastic material. The model takes into account the effect of pre-stress through a consistent linearization, which leads to a more rigorous set of equations than used by previous authors. We have solved the set of linear evolution equations resulting from the model using a series expansion of pressure increment and displacement in terms of scalar and vector spherical harmonics, respectively. For each term of the series, we have obtained an eigenvalue problem

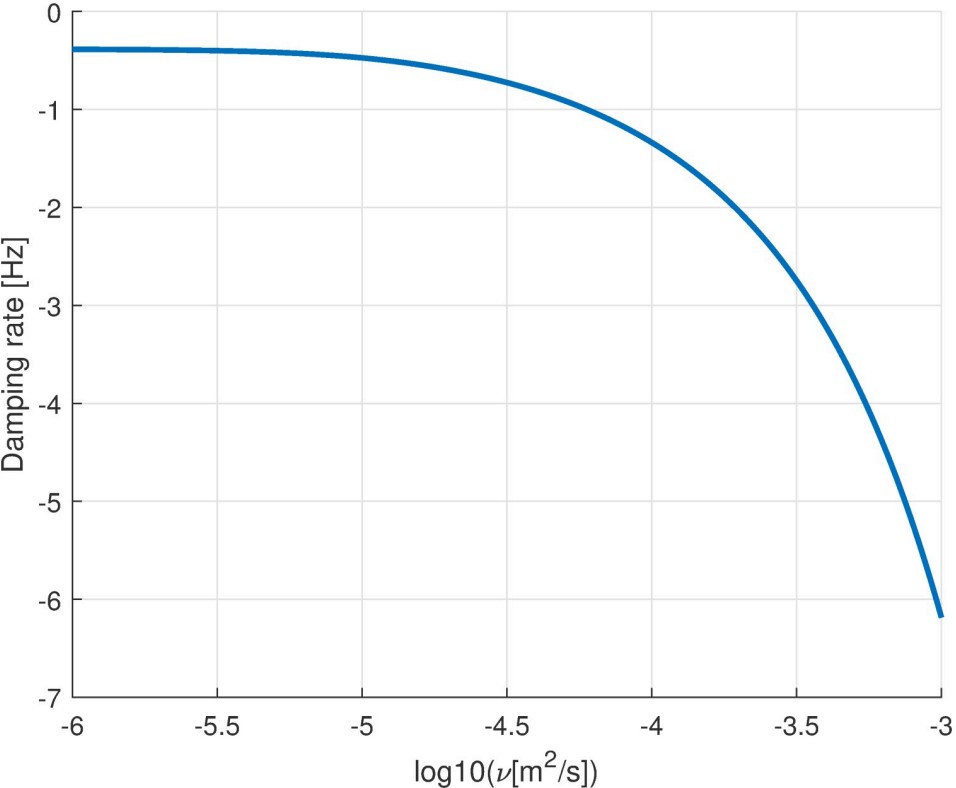

**Fig 7. Damping of the slowest decaying mode as a function of fluid viscosity.**

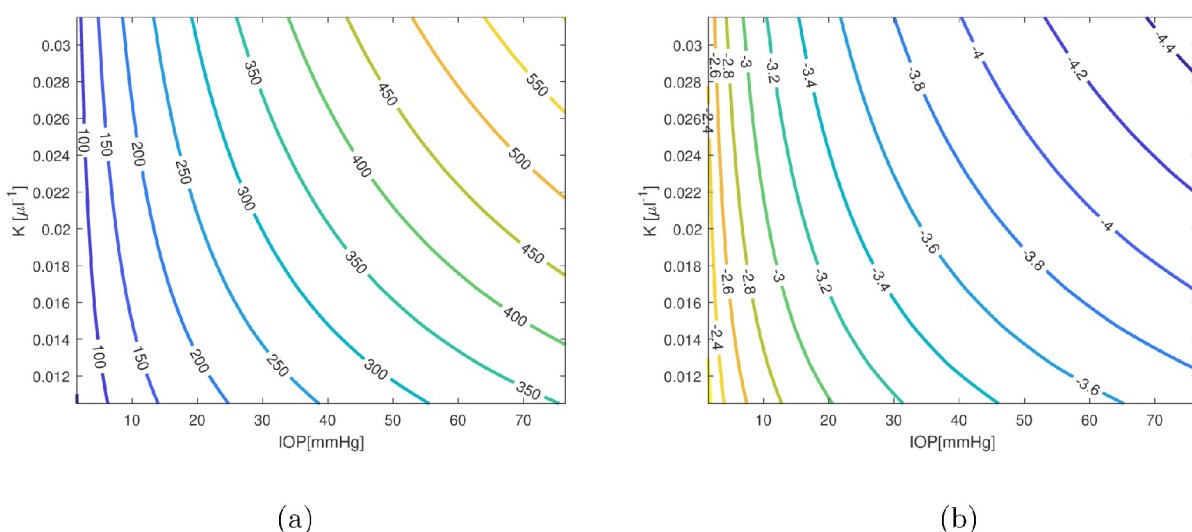

<div align="center">(a)                                                        (b)</div>

**Fig 8. Frequency (a) and damping rate (b) of the slowest decaying mode as a function of IOP and ocular rigidity _K_.**

consisting of three nonlinear algebraic equations, which we have solved numerically. We focused on the bending modes with the lowest damping rate, which are the most relevant for our application.

The study aims to improve our understanding of the mechanical behavior of the eye under dynamic conditions. This is relevant in acoustic tonometry, which is a non-contact technique to measure IOP. The technique is based on the principle that the natural frequencies of oscillation of the eyebulb depend on IOP and it is thus possible to correlate high values of IOP to large oscillation frequencies. Clearly, for this method to be of practical use, one needs to understand what parameters the oscillation frequencies depend on and, in particular, how they vary with IOP. A more comprehensive understanding of the dynamic behavior of the eyebulb can also be relevant for other ocular pathological conditions, such as traumatic retinal detachment.

We have evaluated the natural vibration frequencies and damping rates, using parameters significant for the eye ball dynamics. A novelty with respect to previous works is that we have used ocular stiffness to assess the mechanical properties of the corneo-scleral shell, which we believe is a better approach than the use of a value of Young's module taken from experiments on scleral tissue. This is for two reasons. First, scleral properties are known to be spatially variable, and second, they are significantly different from those of the cornea. The use of a value of Young's modulus based on inflation tests provides a meaningful "average" value. Secondly, the use of Friedenwald's law (11), somehow accounts for the nonlinearity of the corneoscleral tissue.

The existing experimental and numerical estimates of the vibrations of the eyeball provide extremely sparse values (see Table 1 of [33]), ranging from 30 to 800 Hz. This extreme uncertainty probably reflects lack of standardization of the measurement techniques, experimental inaccuracies or difficulty in data interpretation, more than real variability from case to case. Our model shows that the slowest decaying mode of oscillation that involves motion of the eye bulb (and can thus be observed and measured) has frequencies ranging from $\approx 80$ to $\approx 300$ Hz, depending on the value of the IOP. These values are well within the range of measurements. Moreover, the good agreement with the experimental results of Salimi et al. [23], performed on a fluid-filled rubber ball in controlled conditions, is reassuring about the reliability of our predictions.

The increase of the vibration frequencies with IOP is of particular interest in the present context, as it is at the basis of acoustic tonometry. This dependency has two origins: "geometric" stiffening, due to an increase of the pre-stress, and "material" stiffening, due to the nonlinearity of the stress-strain curve of the sclera. Our results indicate that the latter effect is by far dominant, as shown by Fig 6.

Our results also predict a significant dependency of bulb dynamics on the mechanical properties of the corneo-scleral shell and, in particular, on ocular rigidity $K$: the natural frequencies of oscillation of the eye increase with $K$, i. e. with increasing stiffness of the ocular tissues. Fig 8 shows, however, that the dependency of oscillation frequencies on ocular rigidity is important only at relatively large values of IOP and, close to physiological conditions (IOP of $\approx 15$ mmHg), IOP is by far the main determinant of ocular vibration frequencies. This is an important finding, since it confirms that measuring the natural vibration frequencies of the eye is a promising method to measure IOP. In particular, longitudinal measurements on a single subject have the potential to provide reliable estimates of IOP changes.

In this work, we have for the first time considered the effect of the vitreous gel on the dynamics of the eye bulb. Results show that the material filling the eye ball affects significantly the frequencies of oscillation, mostly owing to an added mass effect: an eye filled with gas, which is much lighter than the vitreous, would have higher natural frequencies of oscillation. The material property that matters for the inertia of the system is, obviously, density. On the

other hand, our results show that the elastic and viscous properties of the material filling the vitreous chamber have little effect on ocular vibration frequencies. However, viscosity plays a major role in determining the damping rate of the system. This confirms the speculation that the vitreous body might protect the internal ocular tissues, effectively acting as a damper. This might be particularly relevant in the case of traumas.

The results of our theoretical model confirm that acoustic tonometry is based on sound physical principles and that it is a technique to measure IOP worth being further explored. Specifically, we find that the eyebulb vibration frequencies significantly increase with IOP. Moreover, even if the vibration frequencies also depend on ocular rigidity (which can have an inherent intersubject variability), its effect is significant only at large values of IOP and, for pressures relatively close to physiological, IOP is by far the main determinant of the eyebulb frequencies of oscillation.

Our results may be improved in several aspects. We may take into account the spatial variability of shell properties and the details of eye geometry, which is not exactly a sphere. Accounting for these effects to model the dynamics of the ocular bulb would, however, rule out the possibility of adopting analytical methods and would require a fully numerical approach. As a result, this would make it difficult to capture the role of the key parameters. In this respect, we think that idealized models, such as the one presented here, represent a very valuable complementary approach. This is because analytical models easily elucidate relationships between the quantities at play, allowing to capture the essential features of the problem.

Concerning the clinical relevance our results, we point out that air puff tonometers presently in use are ultimately based on the principle of inducing a deformation of the cornea, through a jet of air and of correlating it to the air pressure. During a typical test the cornea undergoes large deformations so that, at its maximum deflection, its curvature changes sign (from convex to concave). From the mechanical point of view, this is a formidably complex problem, involving nonlinear tissue deformation, fluid motion on the aqueous side, air dynamics on the outer side, with the jet impinging a moving wall. Inertia, due both to the cornea and to the aqueous humor, very likely plays a major role [49], which is not thoroughly understood yet. The mechanics involved in acoustic tonometry is, comparably, much simpler since it involves linear (small amplitude) oscillations of a sphere. In addition, acoustic tonometry is expected to be more suitable for self-administered measurements, offering an easier and more accessible method for individuals to independently monitor their intraocular pressure. We thus think that this technique has the potential to become a robust method to measure IOP. The fact that the existing measurements of ocular natural vibration frequencies are so sparse might seem to be somehow discouraging but, as mentioned earlier, is likely due to a lack of standardization of the procedures.

## Supporting information

**S1 File.**
(PDF)

## Acknowledgments

RR thanks prof. Ross C. Ethier (Georgia Tech., USA) for some useful discussion and for suggesting the use of pressure-volume relationships to estimate the elastic properties of the eyebulb. The authors thank Alessia Ruffini (University of Genoa) for drawing Fig 1.

## Author Contributions

**Conceptualization:** Nicoletta Tambroni, Giuseppe Tomassetti, Rodolfo Repetto.

**Formal analysis:** Nicoletta Tambroni, Giuseppe Tomassetti, Silvia Lombardi.

**Investigation:** Nicoletta Tambroni, Giuseppe Tomassetti, Rodolfo Repetto.

**Software:** Nicoletta Tambroni.

**Supervision:** Rodolfo Repetto.

**Validation:** Silvia Lombardi.

**Writing – original draft:** Nicoletta Tambroni, Giuseppe Tomassetti, Rodolfo Repetto.

**Writing – review & editing:** Nicoletta Tambroni, Giuseppe Tomassetti, Rodolfo Repetto.

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
