## [Decision Letter · Decision Letter 0]

29 Nov 2023

PONE-D-23-36645A mechanical model of ocular bulb vibrations and implications for acoustic tonometryPLOS ONE

Dear Dr. Tomassetti,

Thank you for submitting your manuscript to PLOS ONE. After careful consideration, we feel that it has merit but does not fully meet PLOS ONE’s publication criteria as it currently stands. Therefore, we invite you to submit a revised version of the manuscript that addresses the points raised during the review process.

According to reviewers request please introduce their comments into the final version of the paper.

We look forward to receiving your revised manuscript.

Kind regards,

Pawel Klosowski, D.Sc.

Academic Editor

PLOS ONE

Journal Requirements:

Reviewers' comments:

Reviewer's Responses to Questions

**Comments to the Author**

1. Is the manuscript technically sound, and do the data support the conclusions?

Reviewer #1: Yes

Reviewer #2: Yes

Reviewer #3: Yes

2. Has the statistical analysis been performed appropriately and rigorously? 

Reviewer #1: N/A

Reviewer #2: Yes

Reviewer #3: N/A

3. Have the authors made all data underlying the findings in their manuscript fully available?

Reviewer #1: Yes

Reviewer #2: Yes

Reviewer #3: Yes

4. Is the manuscript presented in an intelligible fashion and written in standard English?

Reviewer #1: Yes

Reviewer #2: Yes

Reviewer #3: Yes

5. Review Comments to the Author

Reviewer #1: The manuscript is written well and easy to follow. Thank to authors of paper to present their great experience on this issue. However, following comments must be addressed:

- The abstract and conclusions is clear but lacks quantitative data to support the statements regarding outcomes of the study, this should be addressed, especially in the conclusion.

- The introduction section needs to add more information on the background and problem statement. Not much has been said about the importance of this study, and this is the weakness of the article.

- Methods were largely rigorous, and appropriate. They were also explained thoroughly.

- The "Conclusions and Future Work" section appears excessively concise, then the authors should expand it by recapping all the steps to the proposed work, as to offer a brief but complete summary of it to the readers.

- How can this research be used in certain clinical applications? Please discuss certain applications or give certain examples.

Reviewer #2: In this paper, a complex mechanical model of ocular bulb vibrations is proposed, which describes the cornea and sclera as a thin prestressed elastic shell, and the vitreous body as a viscoelastic material. The model takes into account the effect of prestressing through sequential linearization. The natural frequencies and the corresponding vibration modes of the system are investigated. It has been demonstrated that the vibration frequency is significantly affected by intraocular pressure (IOP) and the rigidity of the eye, while the rheological properties of the vitreous primarily affect damping.

This study contributes to understanding the mechanical behavior of the eye under dynamic conditions, which could potentially have implications for non-contact methods for measuring intraocular pressure, such as acoustic tonometry. The developed model can also be applied to other ocular pathological conditions, such as traumatic retinal detachment, which are believed to be affected by the dynamic behavior of the eye.

The article is distinguished by a detailed description of the biological, mechanical and mathematical aspects of the problem under consideration, reasoned assumptions. All this makes the text of the interdisciplinary work understandable for specialists in each of these areas. This creates a pleasant impression of the work.

At the same time, there are the following comments on this work:

1. What method was used to solve algebraic eigenvalue problem to determine natural frequencies of vibrations? Why this one was chosen from the many existing methods?

2. There is no information about how numerical results were obtained. Which software packages were used?

3. It is necessary to check and put in order the list of references because the following articles are duplicated: No. 19 and No. 30, No. 20 and No. 32, No. 21 and No. 36.

4. Not all quantities and variables used in equations are explained in the text. For example, the parameter � in formula (5) is not defined in the article.

5. On page 8 in line 254 (in the paragraph before eq. (25)) there is a reference to the equation of motion, but the equation number is indicated as (??).

The article can be recommended for publication in the journal «PLOS ONE» with minor revision.

Reviewer #3: The article is interesting both theoretically (construction of a biomechanical model of the eyeball) and in terms of the prospects for practical implementation of the model in acoustic tonometry technology.

The authors have solved, for the first time, the set of linear evolution equations resulting from the model using a series expansion of pressure increment and displacement in terms of scalar and vector spherical harmonics.

An important advantage of the proposed model is that it takes into account the nonlinear dependence of the corneoscleral rigidity on intraocular pressure. Taking account of the physical characteristics of the vitreous body and their impact on the resulting values of vibration frequencies is a new and significant result, too.

I have no major comments. It would be good to see the authors’ opinion on the following issue: the results of all known tonometry methods, except for intraocular pressure measurement using the Ocular Response analyzer (ORA), are influenced by the biomechanical properties of eye shells. Are there any advantages in this regard in acoustic tonometry, which can be based on the proposed model? As shown in the work, vibration frequencies and damping rates are affected by both IOP and the mechanical properties of the corneoscleral shells. Then what is the advantage of acoustic tonometry other than non-contact measurement mode?

6. PLOS authors have the option to publish the peer review history of their article (what does this mean?). If published, this will include your full peer review and any attached files.

Reviewer #1: No

Reviewer #2: No

Reviewer #3: No

---

## [Author Response · Author response to Decision Letter 0]

12 Dec 2023

We thank the Referees for the time and effort spent

reviewing our work. Our reply is in the enclosed letter.

---

## [Editor Report · Decision Letter 1]

14 Dec 2023

A mechanical model of ocular bulb vibrations and implications for acoustic tonometry

PONE-D-23-36645R1

Dear Dr. Tomassetti,

We’re pleased to inform you that your manuscript has been judged scientifically suitable for publication and will be formally accepted for publication once it meets all outstanding technical requirements.

Kind regards,

Pawel Klosowski, D.Sc.

Academic Editor

PLOS ONE

Additional Editor Comments (optional):

The reviewers required minor revisions to the previous version. The Authors have made necessary changes.  To accelerate the review process I decide to accept the paper in the current version.
---

## [Editor Report · Acceptance letter]

8 Jan 2024

PONE-D-23-36645R1 

PLOS ONE

Dear Dr. Tomassetti, 

I'm pleased to inform you that your manuscript has been deemed suitable for publication in PLOS ONE. Congratulations! Your manuscript is now being handed over to our production team.

Kind regards, 

on behalf of

Prof. Pawel Klosowski 

Academic Editor

PLOS ONE